# Time course and heterogeneity of treatment effect of the collaborative chronic care model on psychiatric hospitalization rates: A survival analysis using routinely collected electronic medical records

**Michael A. Ruderman**[1,2¤a¤b]*, **Bo Kim**[1,2], **Kelly Stolzmann**[1], **Samantha Connolly**[1,2], **Christopher J. Miller**[1,2], **Mark S. Bauer**[1,2]

**1** Center for Healthcare Organization & Implementation Research, VA Boston Healthcare System, Jamaica Plain, Boston, Massachusetts, United States of America, **2** Department of Psychiatry, Harvard Medical School, Boston, Massachusetts, United States of America

¤a Current address: Division of Mental Health Services, San Francisco Veterans Affairs Medical Center, San Francisco, California, United States of America
¤b Current address: Department of Psychiatry, University of California, San Francisco, California, United States of America

* mikeruderman@gmail.com

**Data Availability Statement:** The Office of Research Oversight (ORO) in the Department of

## Abstract

### Background

Health systems are undergoing widespread adoption of the collaborative chronic care model (CCM). Care structured around the CCM may reduce costly psychiatric hospitalizations. Little is known, however, about the time course or heterogeneity of treatment effects (HTE) for CCM on psychiatric hospitalization.

### Rationale

Assessment of CCM implementation support on psychiatric hospitalization might be more efficient if the timing were informed by an expected time course. Further, understanding HTE could help determine who should be referred for intervention.

### Objectives

(i) Estimate the trajectory of CCM effect on psychiatric hospitalization rates. (ii) Explore HTE for CCM across demographic and clinical characteristics.

### Methods

Data from a stepped wedge CCM implementation trial were reanalyzed using 5 570 patients in CCM treatment and 46 443 patients receiving usual care. Time-to-event data was constructed from routine medical records. Effect trajectory of CCM on psychiatric hospitalization was simulated from an extended Cox model over one year of implementation support. Covariate risk contributions were estimated from subset stratified Cox models without using

Veterans Affairs Veterans (VA) Health Administration provides guidance for VA researchers. The purview of ORO includes the dissemination of VA research findings and associated final datasets to the public. To protect subjects, ORO directs investigators to obtain written agreements, whenever feasible, from members of the public agreeing that they will not attempt to identify or re-identify subjects from a final de-identified anonymous dataset. Institutional Review Board and VA Research and Development Committee approvals are required for outside entities to conduct research using VA patient data. Interested persons can find more information to initiate this process by visiting https://www.virec.research.va.gov or contacting the VA Information Resource Center at VIReC@va.gov, and the study team at the VA Center for Healthcare Organization and Implementation Research can be reached by emailing CHOIR@va.gov.

**Funding:** BK, KS, SC, CJM, MAR, and MSB acknowledge financial support from the United States Department of Veterans Affairs, Office of Mental Health and Suicide Research via grants through their Quality Enhancement Research Initiative, grants QUE-15-289 and CIN-13-403 (url: https://www.research.va.gov/). This funder was a partner in designing and conducting the primary trial on which the current study is based. This funder had no role in the analysis, decision to publish, or preparation of the manuscript for the current study. MAR acknowledges receiving statistical consultation financially supported by the Harvard Catalyst | The Harvard Clinical and Translational Science Center (National Center for Advancing Translational Sciences, National Institutes of Health Award UL 1TR002541) and financial contributions from Harvard University and its affiliated academic healthcare centers. The content is solely the responsibility of the authors and does not necessarily represent the official views of Harvard Catalyst, Harvard University and its affiliated academic healthcare centers, or the National Institutes of Health (url: https://catalyst.harvard.edu/services/biostatsconsult/). This funder had no role in study design, data collection, and analysis (beyond initial consultation), decision to publish, or preparation of the manuscript. MAR received salary support from the Harvard South Shore Psychiatry Residency Training Program (url: https://www.harvardsouthshore.org/research.html). This funder had no role in the design, data collection, analysis, decision to publish, or preparation of the manuscript for the current study.

**Competing interests:** The authors have declared that no competing interests exist.

simulation. Ratios of hazard ratios (RHR) allowed comparison by trial arm for HTE analysis, also without simulation. No standard Cox proportional hazards models were used for either estimating the time-course or heterogeneity of treatment effect.

## Results

The effect of CCM implementation support increased most rapidly immediately after implementation start and grew more gradually throughout the rest of the study. On the final study day, psychiatric hospitalization rates in the treatment arm were 17% to 49% times lower than controls, with adjustment for all model covariates (HR 0.66; 95% CI 0.51–0.83). Our analysis of HTE favored usual care for those with a history of prior psychiatric hospitalization (RHR 4.92; 95% CI 3.15–7.7) but favored CCM for those with depression (RHR 0.61; 95% CI: 0.41–0.91). Having a single medical diagnosis, compared to having none, favored CCM (RHR 0.52; 95% CI 0.31–0.86).

## Conclusion

Reduction of psychiatric hospitalization is evident immediately after start of CCM implementation support, but assessments may be better timed once the effect size begins to stabilize, which may be as early as six months. HTE findings for CCM can guide future research on utility of CCM in specific populations.

## Introduction

### Background rationale

Healthcare organizations increasingly are adopting collaborative chronic care models (CCMs) to organize team-based clinic staff, unify evidence-based standards, enhance program evaluation, and make connections to broader community resources. The CCM was developed to improve management of chronic health conditions by delivering multi-component care in outpatient primary care settings. This model is a departure from traditional individual clinical practices, which were often siloed from other specialties and broader treatment goals [1–3]. CCMs consist of six elements, each flexibly implemented depending on local priorities and needs [1,4,5]. These elements include work role redesign for anticipatory, continuous care; patient self-management support; provider decision support; information management; linkage to community resources and leadership as well as organizational support. CCMs now have a strong evidence base for mental health care, including depression treated in primary care and persistent mental health conditions treated in mental health clinics [4,5].

For mental health diagnoses specifically, meta-analyses suggest that CCM-based care is associated with improved clinical outcomes, quality of life, social role function, and global mental health [4,5]. However, these trials typically do not reflect real-world conditions, excluding patients whom are unwilling, legally incompetent, or are otherwise unable to provide informed consent. This could eliminate patients with unidentified factors that potentially interact with specific diagnoses or comorbidities (positively or negatively) and bias results.

Ultimately, any cost-effective or sustainable CCM implementation support will require an accounting of real-world conditions and patient factors. Knowing the trajectory of an intervention enables understanding when deviations from the effect's expected decay or growth are

likely to occur. This can inform the optimal timing for conducting an evaluation to detect such deviations, and thus minimize redundant evaluations and their associated costs.

Significantly, a recent implementation trial, under real-world conditions, in US Department of Veterans Affairs (**VA**) medical centers showed that one year of CCM implementation support decreased psychiatric hospitalization rates in general mental health clinics versus general mental health clinics from the same medical centers that did not receive CCM implementation support. The current report utilizes data from this primary study to identify demographic and clinical characteristics associated with reduction in hospitalization rates. It uses survival analysis, allowing for control of multilevel covariates, including individual characteristics included in the primary study analysis, while providing fine temporal resolution of the time course for effect attributable to CCM implementation support.

This study is an exploratory analysis of two research questions, (i) what is the time course of CCM's effect on psychiatric hospitalization? And (ii) what is the heterogeneity of its treatment effect across demographic and clinical characteristics? Study results will have implications for how to deploy the CCM in real-world practice—e.g., whether there are specific subgroups that might be targeted for treatment with the CCM, and whether the clinical effects reach steady state over the year of implementation support.

## Methods

### Primary study overview

This study is a new analysis of a randomized implementation trial that utilized a stepped wedge design. The initial study followed a hybrid type II design, examining both implementation fidelity and feasibility, as well as health outcome effectiveness, including psychiatric hospitalization. Details about implementation aspects of the primary study, including quantitative measures, nature of facilitated implementation support, site selection, and roll-out of the stepped wedge design have been previously published, with the trial protocol also available [6,7]. A summary of the protocol follows.

### Ethics statement

This study was approved by the VA Central Institutional Review Board (IRB) as a combined quality improvement (QI) program evaluation and research project. Implementation efforts and related outcome measures (those routinely being collected by the health care system for its operational purposes—e.g., demographic and clinical characteristics, including hospitalizations) were considered to be for program evaluation of mental health care quality improvement, and were thus determined to be exempt from IRB review. Given this exemption, informed consent was not required to access patient records used in the study, which were not anonymized before they were accessed (only by approved project personnel).

### Setting and intervention in the primary study trial

The primary study trial sites were recruited from all VA medical centers via top-down recruitment through national and regional conference calls. Sites were asked to: (a) provide four hours per week of an internal facilitator with mental health process improvement experience from existing staff and (b) identify one clinic from among their general mental health clinics to implement the CCM including allocating one hour per week for process improvement meetings. The first sites to complete a project letter of agreement were entered into the trial. All nine VA facilities included in the initial study were included in this new analysis. Following a stepped wedge design, CCM implementation support commenced in three randomized

waves spaced by four months each, with the first wave starting February 2016. Randomization was performed using a novel approach termed sequential balance [8].

The intervention consisted of one year of implementation support provided by three external facilitators assigned to three sites each, and who were expert in both CCM care and implementation facilitation [9]. The external facilitator was funded by the study and worked with the internal facilitator and CCM team to review clinical processes and revise them to align with CCM principles. The external facilitator worked primarily remotely with the facility, using video and conference calls, email, and electronic document sharing. The implementation process was guided by a step-by-step workbook developed for the primary study (available on request).

## Participants

All patients who were active in the CCM clinics at the start of implementation support were included as the study population, including all diagnostic groups excluding individuals with dementia. Patients were considered active in the clinic if they had had at least two visits to the CCM clinic within the year prior to implementation support, including one visit in the quarter before the start of implementation. Consistent with an intent-to-treat approach, all qualifying patients were included in the outcome analyses regardless of their subsequent involvement in the clinic. For the hospitalization analyses, the CCM group was compared with a control group, consisting of all active patients in all other general mental health clinics in the same medical centers utilizing the same exclusion and inclusion criteria.

## Variables

While the initial study investigated overall one-year hospitalization rates, for the current analyses the dependent variable is time until psychiatric hospitalization (measured as days since implementation start date) where a 'psychiatric hospitalization' refers to any admission to an acute level inpatient psychiatric unit.

**Independent variables.** Variables included age (grouped by decade), sex, minority status (defined as race other than "White" or ethnicity of "Latino"), marital status (married, or any status other than married), rural residence (urban or either, rural or highly rural), military service period (Gulf War and before, or post-Gulf War), disability connected to military service ($\geq$50% disability level vs. lesser or no disability level), psychiatric diagnoses, psychiatric hospitalization in the year prior to implementation, number of psychiatric comorbidities (two or fewer diagnoses, three or more diagnoses), number of medical comorbidities (none, one, or multiple medical diagnoses), and the clinic site at which the patient received care.

## Data sources

All variables were extracted from electronic health record (**EHR**) data hosted by VA's Corporate Data Warehouse (**CDW**) [10]. Every patient had individual-level data entered into the EHR as part of their routine clinical care over the study's course. The time-to-event variable (time until acute psychiatric hospitalization) was defined by subtracting the date of a patient's admission, as recorded in the EHR, from the date corresponding that patient's study start date, defined as their clinic's protocol implementation start date.

**Data access and cleaning methods.** Time-to-event data was constructed from dates of admission in CDW for all patients included in the study sample. Primary tasks of data cleaning included identifying scenarios in which patients were either censored, admitted, or died either before, or during the study period. Once identified, data were programmed into a counting-process data format for preliminary assessment of recurring events (readmissions) as well as a

time-varying extended Cox model. This format splits observation time intervals for each individual patient into discrete periods. This was done in two steps. First, intervals were split into discrete periods between any *repeat* hospitalization (readmissions) using the *Survival* software Package for the R statistical environment [11]. Second, intervals were further subdivided into discrete periods so as to align with *any other psychiatric hospitalizations across the entire dataset*; this second step was accomplished with the *SimPH* statistical package in R [12].

## Statistical methods

**Modeling and analytic overview.**   Our modeling process utilized the collective expertise of VA's Center for Healthcare Organization & Implementation Research. Initial model specification relied on iterative discussions between team members on potential confounders, colliders, or other sources of bias. Our choice among the candidate models was guided by information theory, relying on relative scores after applying Akaike information criterion.

We used survival analysis to explore time course and heterogeneity of CCM's effect on psychiatric hospitalization across demographic, clinical, and site level characteristics, as well as the overall variation of treatment effect over time. Adjusted hazard ratios (**HRs**) and confidence intervals (**CIs**) were visualized continuously over time by graphing multiple HR point estimates simulated from an extended Cox model for each day. To provide single point estimates of covariate risk contributions, by trial arm, we used stratified Cox models subset by trial arm. To describe heterogeneity of treatment effect (**HTE**), we used a second differences approach by calculating ratios of hazard ratios (**RHR**) for each covariate.

**Descriptive statistics.**   Baseline characteristics by trial arm were compared by multiple Chi-square tests, and effect sizes estimated by Cramer's V to accommodate multi-level variables.

## Missing data

Our imputation strategy used a classification and regression trees method, as implemented in the *mice* package for the R statistical environment [13].

## Handling of time-to-event data

The final dataset was left-truncated and handled through adjustment for age. Some patients had multiple hospitalizations during the study period. A preliminary model included recurrent data, with adjustment for intra-patient correlation by use of a robust estimator. However, the resulting model did not substantially improve power or alter the main effect. Given this, we did not include recurrent event data in any of our final models.

## Diagnostic testing and choice of Cox proportional hazards model variants

The Cox proportional hazards model assumes a constant effect size over time, known as the proportional hazards assumption (**PHA**). This includes all variables specified in the model, including trial status. We tested the PHA in two ways, analytically and visually. First, we regressed scaled Schoenfeld residuals plotted over log-time to detect no zero slopes using a conventional 95 percent threshold. Second, all plots were visually inspected to detect non-monotonic patterns missed by regression.

Variables that violated the PHA were specified as functions of log-transformed time in the extended Cox model. Simulations from this extended variant of the Cox model aided visualization of the relative effect of CCM versus controls over time. However, we required additional models to assess HTE due to the trials status variable itself violating the PHA. Therefore, we

used two additional models subset by treatment status, effectively removing the trial status variable from both models. We chose stratified Cox models for each subset to account for other variables that violated the PHA even after subsetting. The resulting two subset stratified models provided the relative effect measures of covariates and were used to calculate our HTE measures, the RHR.

**CCM effect estimation.** We estimated CCM's effect on psychiatric hospitalization as a function of log-time using an extension of the Cox model that handles time-varying, or time-interactive, effects. Additionally, any covariates violating the PHA were also specified as functions of log-time.

## Visualization of time-varying effects

The extended Cox model was used to generate post-estimate simulations to clearly visualize time-interactive properties and variance of the main CCM effect (i.e., as instantaneous hazard ratios over the course of study). For each study day, we simulated 1 000 'draws' from a probability distribution corresponding to the variance-covariance matrix of the underlying model. The results of this post-estimation simulation were plotted as adjusted HR point estimates, where CIs are defined by the regions bounding 95% of the point estimates. We used the *simPH software* package for the R statistical environment [14] to perform our post-estimation simulations.

## Covariate risk contributions by trial arm

Individual covariate risk contributions could not be estimated by a single standard Cox model because the trial status variable violated the PHA. Instead, we created two identically specified Cox models that were subset by trial arm (thus removing the time-varying trial status variable). Both subset models required a stratified Cox procedure because other covariates were discovered to violate the PHA even after subsetting out trial status. The PHA was satisfied for the resulting non-standard stratified Cox models. Importantly, while this approach does not provide point estimates for PHA violators, it does adjust for their effects in the point estimates of the remaining covariates.

**Heterogeneity of treatment effect.** Our approach to analyzing HTE is essentially a second differences approach generalized for probability ratios [15,16]. We started with the stratified Cox models subset by trial arm, as described above, and divided the HRs from the CCM subset model over the HRs from the control subset model. The resulting quotients gave us point estimates of second differences, the RHRs. We also estimated 95% CI for each RHR.

*Sensitivity analysis*. We used the E-value to quantify unmeasured confounding without assuming specific associations between exposure, outcome, and an unknown latent variable. Specifically, the E-value is the minimal magnitude of unmeasured confounding (and independent of measured confounding) that is needed to explain away study results [17].

## Results

### Participants

In all, 5 570 patients from the nine sites were followed as the experimental CCM implementation support trial arm, and 46 443 patients were followed under a control trial arm. These counts were fewer than the 5 596 CCM patients and 46 755 control patients included in the primary study [18]. The difference is accounted for by deaths that occurred after inclusion in the study population, but prior to CCM implementation support start dates. Both the CCM

and control cohorts again represent the entire active clinic populations (excluding patients with dementia).

## Descriptive data

**Patient characteristics.** As expected, patient demographic and clinical characteristics (Table 1) were not substantively different across treatment arms, and resembled the distributions reported in the primary study [18]. Specifically, due to the large sample sizes, most variables were significantly different across groups; however, effect sizes were small by Cramer V estimates. To the extent residual differences exist, we have controlled for all given covariates in our effect modeling.

## Follow-up time

Censoring due to death was accounted for through CDW data that is informed and updated by multiple national death registries. There were 982 deaths across both trial arms during the one-year study observation period.

**Missing data.** Prior to imputation, most data for covariates had no missing values. We attributed the high level of completeness for this large dataset to the fact that it was collected as a required part of routine care; those that did have missing values were well under five percent.

## Outcome data

Over the study year, there were 2 364 admissions recorded for all 52 013 patients in our study. These admissions were connected to 1 520 unique patients, with the surplus attributable to patients with multiple hospitalizations. The maximum number of admissions experienced by a single patient over the study was 10 in the CCM cohort and 14 in the control cohort.

## Main results

**Time-varying CCM effect on psychiatric hospitalization.** Fig 1 shows the time-varying effect of CCM. At baseline, the rate of hospitalization for CCM was nearly one and a half times greater than controls on day one (HR 1.48; 95% CI 0.79–1.75). However, this difference rapidly neutralized by day 16 of CCM implementation support (HR 1.0; 95% CI 0.77–1.28). By day 56, the CI's upper-bound crossed unity. The magnitude of CCM's effect continued to grow (decreasing HR) throughout the study. And the rate at which the magnitude grew decelerated throughout the same period. Notably, on the last study day, visual inspection of Fig 1 illustrated a slope that was still non-zero. The maximum CCM effect observed was on the final study day, where psychiatric hospitalization rates for patients in the CCM cohort were estimated between 17% to 49% times lower than controls, with adjustment for all model covariates (HR 0.66; 95% CI 0.51–0.83) (Fig 1). The calculated E-value for this effect in our sensitivity analysis was 2.4, and for the upper bound of its 95% CI, the E-value was 1.7. E-value results are only interpretable in context (please see discussion subsection on unmeasured confounding).

**Covariate risk contribution results.** Estimated HRs with CIs for adjusted covariate risk contributions are visualized by trial arm strata in Fig 2. The solid dots represent HR point estimates for each stratum, and the width of the 'wings' represent the ranges of their CI. These results are precursors to the RHR, but also provide information about the direction of effects (i.e., HRs less than or greater than one) which are not evident as second differences. Similarly, significant effects experienced relatively equally by both strata were readily appreciated in Fig 2, but not in our HTE analysis. Specifically: schizophrenia, substance use disorders, prior psychiatric hospitalization, and military service after the Gulf War were risk factors for psychiatric

**Table 1. Baseline characteristics and descriptive statistics by trial arm.**

| Variable [a] | No. (%) [b] | | χ2 Test Statistic | P-value | Cramer V Effect Size |
|---|---|---|---|---|---|
| | Controls | CCM | | | |
| | (n = 46,443) | (n = 5,570) | | | |
| Age group (by years): | | | 62 | < 0.0001 | 0.0345 |
| 18–28 | 2,056 (4) | 256 (5) | | | |
| 29–38 | 8,051 (17) | 1,015 (18) | | | |
| 39–48 | 7,034 (15) | 982 (18) | | | |
| 49–58 | 9,247 (20) | 1,204 (22) | | | |
| 59–68 | 12,816 (28) | 1,360 (24) | | | |
| ≥ 69 | 7,239 (16) | 753 (14) | | | |
| Female | 6,388 (14) | 879 (16) | 17 | < 0.0001 | 0.0181 |
| Minority (racial/ethnic) | 15,259 (33) | 1,717 (31) | 9.32 | 0.002 | 0.0134 |
| Married | 22,271 (48) | 2,613 (47) | 2.16 | 0.14 | 0.0065 |
| Rural residence | 14,811 (32) | 1,199 (22) | 251 | < 0.0001 | 0.0694 |
| Post-Gulf War service | 22,195 (48) | 2,856 (51) | 24.2 | < 0.0001 | 0.0216 |
| ≥ 50% disabled [c] | 35,964 (77) | 4,322 (78) | 0.07 | 0.8 | 0.0012 |
| Depression | 17,403 (37) | 2,472 (44) | 101 | < 0.0001 | 0.0440 |
| Anxiety | 13,240 (29) | 1,758 (32) | 22.6 | < 0.0001 | 0.0209 |
| Bipolar | 9,426 (20) | 1,659 (30) | 267 | < 0.0001 | 0.0717 |
| Schizophrenia | 2,402 (5) | 394 (7) | 35.4 | < 0.0001 | 0.0261 |
| Personality disorder | 1,533 (3) | 282 (5) | 45.9 | < 0.0001 | 0.0297 |
| PTSD | 20,810 (45) | 2,781 (50) | 52.6 | < 0.0001 | 0.0318 |
| Substance use diagnosis | 9,917 (21) | 1,090 (20) | 9.49 | 0.002 | 0.0135 |
| Prior psych hospitalization | 1,599 (3) | 245 (4) | 13.3 | 0.0004 | 0.0411 |
| Over 3 psych diagnoses | 7,706 (17) | 1,203 (22) | 87.8 | < 0.0001 | 0.0160 |
| Medical morbidity | 1.98 | 0.37 | 0.0062 | | |
| None | 23,148 (50) | 2,831 (51) | | | |
| Single | 11,336 (24) | 1,339 (24) | | | |
| Multiple | 11,959 (26) | 1,400 (25) | | | |
| Site: | | | 1490 | < 0.0001 | 0.1693 |
| Clinic A | 4,770 (10) | 754 (14) | | | |
| Clinic B | 11,922 (26) | 933 (17) | | | |
| Clinic C | 5,007 (11) | 596 (11) | | | |
| Clinic D | 4,702 (10) | 473 (8) | | | |
| Clinic E | 2,740 (6) | 948 (17) | | | |
| Clinic F | 4,136 (9) | 310 (6) | | | |
| Clinic G | 2,104 (5) | 524 (9) | | | |
| Clinic H | 4,913 (11) | 576 (10) | | | |
| Clinic I | 6,149 (13) | 456 (8) | | | |

[a] Abbreviations: PTSD, Post-traumatic stress disorder; Psych, Psychiatric; Minority, (race other than 'white' and ethnicity other than 'Latinx').

[b] Compared to the total counts initially reported for this trial [18] any differences are accounted for by patients who died during the waiting period, but before start of CCM implementation.

[c] Service-connected disability scored on a scale of (0–100%) and reflects the degree of functional impairment and determines conditions of eligibility of VA's services and supplementary income.

hospitalization in both CCM and control cohorts. And clinics C and G were both protective factors against psychiatric hospitalization in both cohorts. Main effect sizes were overall robust when we excluded non-significant variables or addition of select covariate interactions (which

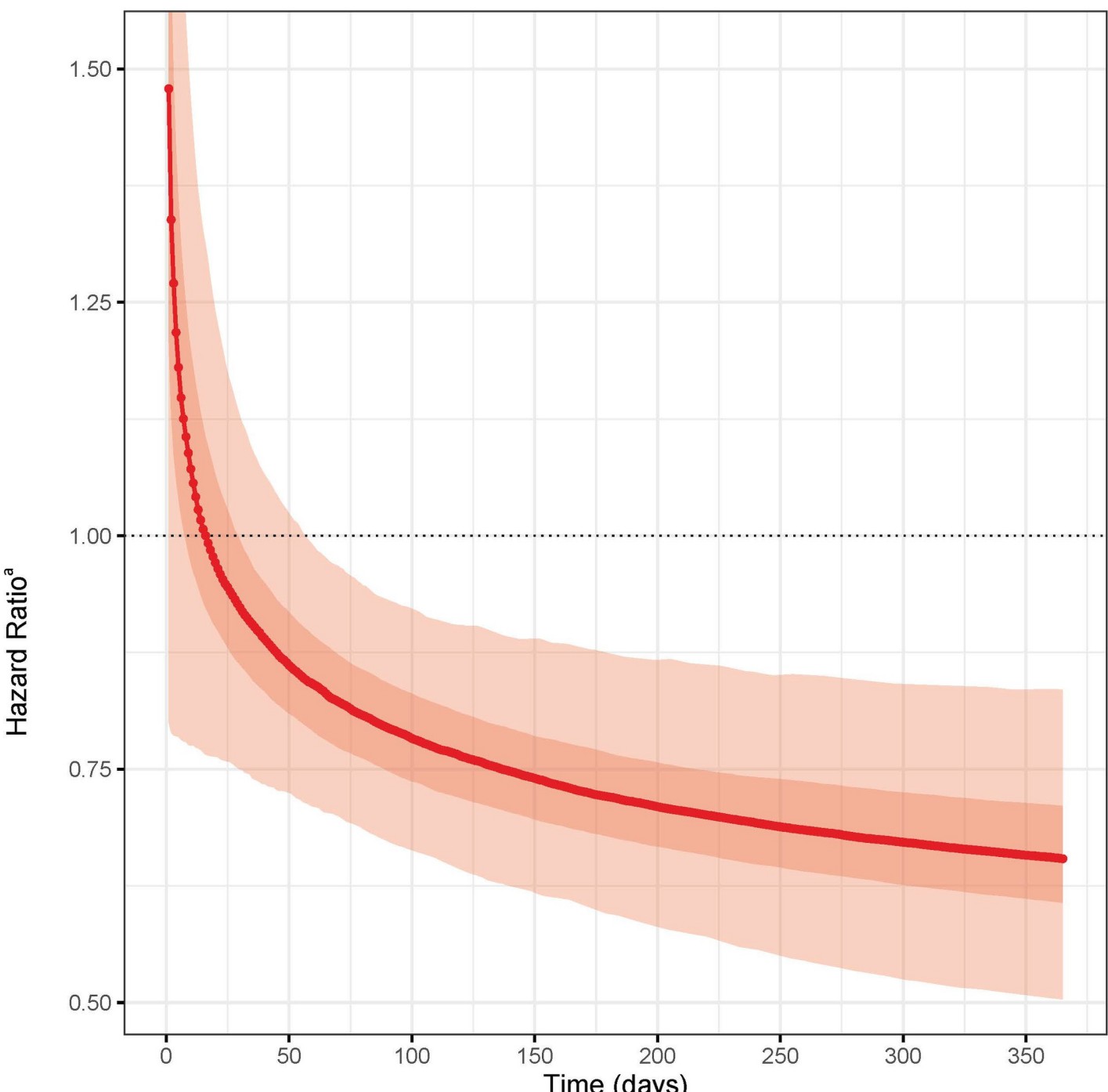

**Fig 1. Adjusted time-varying effect of CCM on psychiatric hospitalization compared to controls (simulated hazard ratio point estimates and 95% confidence intervals for each day since start of trial).** [a] The red line here represents the point estimates for the hazard ratio over time. The surrounding red shading spans the 95% confidence interval for these estimates, with the inner dark red shading showing a single standard deviation.

were not statistically significant). All estimates are adjusted for other model covariates, including those violating the PHA (though estimates for violators are not themselves explicitly modeled). Importantly, these estimates are not derived from the extended model used to simulate time-varying effects in Fig 1.

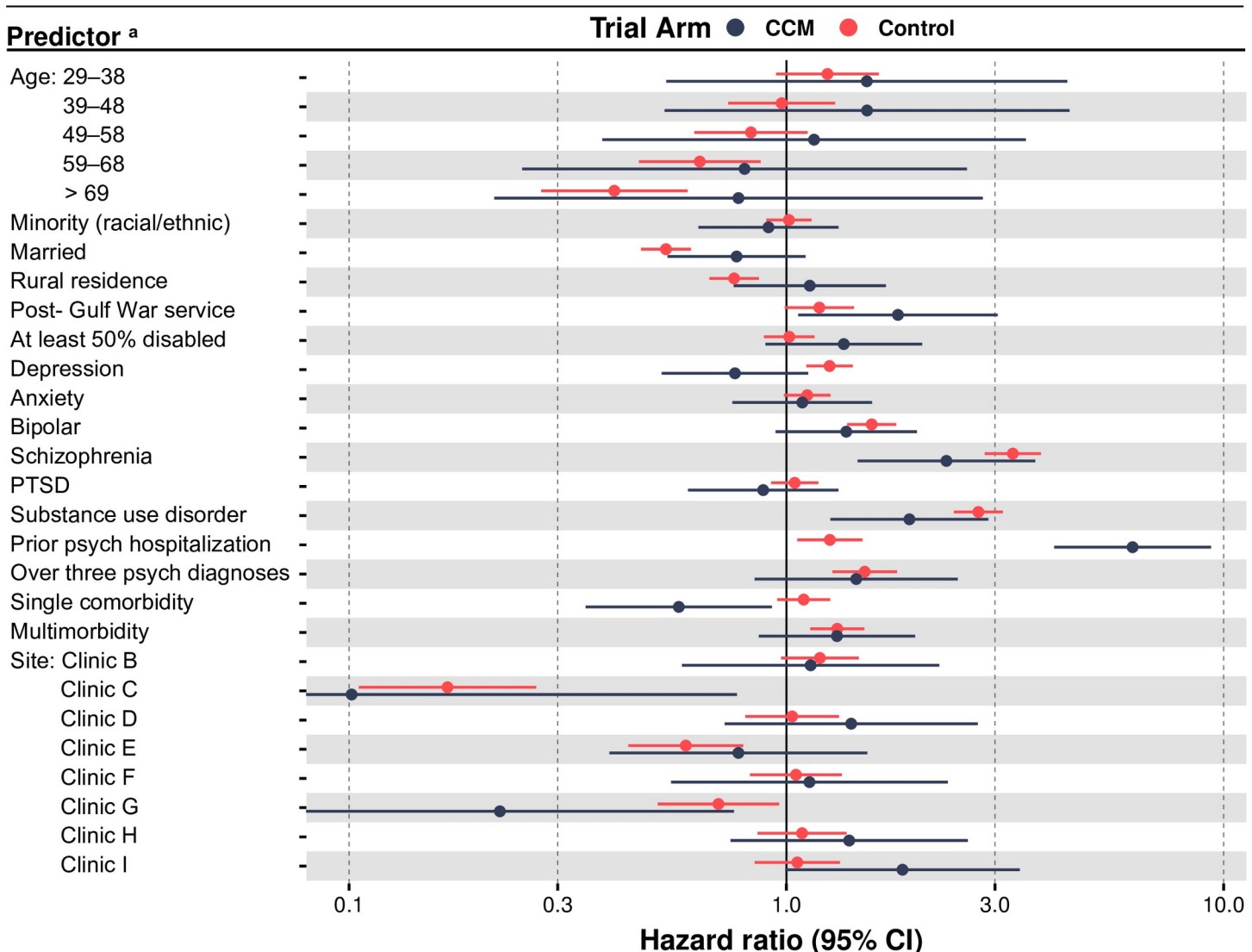

**Fig 2. Adjusted individual risk contributions of predictors for psychiatric hospitalization subset by trial arm (hazard ratios with 95% confidence intervals).** [a] Risk contributions are estimated with a stratified Cox procedure due to violation of the proportional hazards assumption for two independent variables (sex and personality disorder). While all other covariates in the model reflect adjustment for both sex and personality disorder, their direct effects cannot be estimated and are hence excluded from the forest plot.

**Heterogeneity of treatment effects.** We used RHRs with 95% CIs to provide a second differences measure of HTE across covariate subgroups. Fig 3 illustrates the RHRs for all covariates in a forest plot (except for variables violating the PHA). RHRs less than 1.0 suggest greater reduction in psychiatric hospitalization among the CCM cohort, while RHR values greater than 1.0 support greater reductions for the control cohort. Among psychiatric diagnoses, only depression was associated with CCM reducing psychiatric hospitalization rates (RHR 0.61; 95% CI: 0.41–0.91). General medical comorbidity demonstrated an inverted U pattern. Having a single medical diagnosis, compared to having none, favored CCM for reducing psychiatric hospitalization (RHR 0.52; 95% CI 0.31–0.86). However, comparing those with any multimorbidity (two or more medical conditions) with those without any medical condition failed to show any preference for CCM or controls. Overall, the strongest evidence for HTE was among

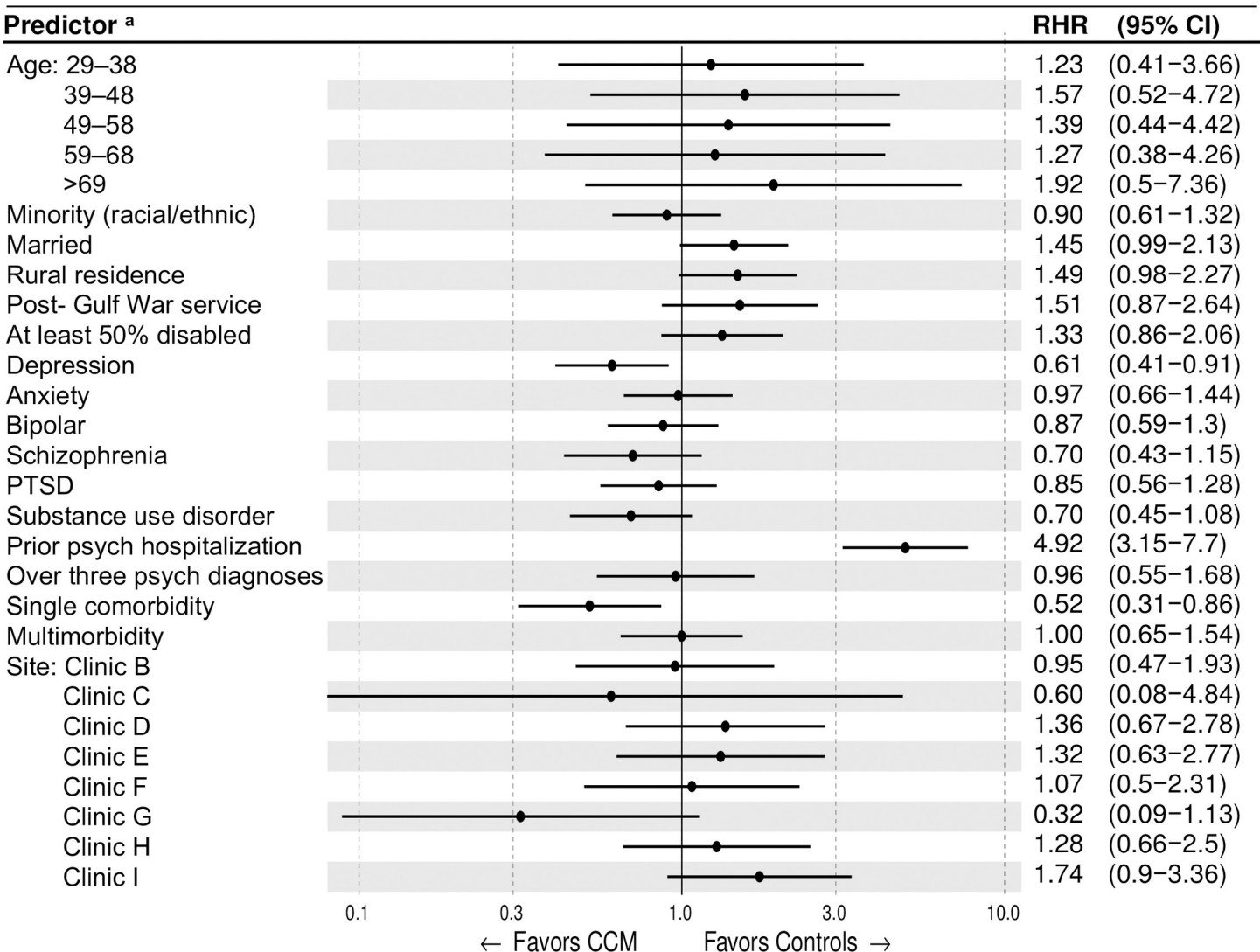

| Predictor [a] | RHR | (95% CI) |
|---|---|---|
| Age: 29–38 | 1.23 | (0.41−3.66) |
| 39–48 | 1.57 | (0.52−4.72) |
| 49–58 | 1.39 | (0.44−4.42) |
| 59–68 | 1.27 | (0.38−4.26) |
| >69 | 1.92 | (0.5−7.36) |
| Minority (racial/ethnic) | 0.90 | (0.61−1.32) |
| Married | 1.45 | (0.99−2.13) |
| Rural residence | 1.49 | (0.98−2.27) |
| Post- Gulf War service | 1.51 | (0.87−2.64) |
| At least 50% disabled | 1.33 | (0.86−2.06) |
| Depression | 0.61 | (0.41−0.91) |
| Anxiety | 0.97 | (0.66−1.44) |
| Bipolar | 0.87 | (0.59−1.3) |
| Schizophrenia | 0.70 | (0.43−1.15) |
| PTSD | 0.85 | (0.56−1.28) |
| Substance use disorder | 0.70 | (0.45−1.08) |
| Prior psych hospitalization | 4.92 | (3.15−7.7) |
| Over three psych diagnoses | 0.96 | (0.55−1.68) |
| Single comorbidity | 0.52 | (0.31−0.86) |
| Multimorbidity | 1.00 | (0.65−1.54) |
| Site: Clinic B | 0.95 | (0.47−1.93) |
| Clinic C | 0.60 | (0.08−4.84) |
| Clinic D | 1.36 | (0.67−2.78) |
| Clinic E | 1.32 | (0.63−2.77) |
| Clinic F | 1.07 | (0.5−2.31) |
| Clinic G | 0.32 | (0.09−1.13) |
| Clinic H | 1.28 | (0.66−2.5) |
| Clinic I | 1.74 | (0.9−3.36) |

0.1        0.3        1.0        3.0        10.0

← Favors CCM        Favors Controls →

**Fig 3. Heterogeneity of CCM treatment effects by individual predictors of psychiatric hospitalization (adjusted ratio of hazard ratios [RHR] with 95% confidence intervals).** [a]RHRs were estimated with a stratified Cox procedure due to violation of the proportional hazards assumption for two independent variables (sex and personality disorder). While all other covariates in the model reflect adjustment for both sex and personality disorder, their direct effects cannot be estimated and are hence excluded from the forest plot.

those with a prior psychiatric hospitalization in the previous year, favoring the control condition with preferential reduction in hospitalization by a factor of almost five (RHR 4.92; 95% CI 3.15–7.7).

## Discussion

### Key results

CCM was associated with a reduced rate of psychiatric hospitalization over our one-year study period. The effect began changing immediately upon start of CCM implementation support, with the fastest change in effect magnitude occurring within the first month. The effect size grew through the last observation day, where psychiatric hospitalization rates were 17% to 49% times lower than controls. Independent of trial arm, schizophrenia, substance use

disorders, prior psychiatric hospitalization, and post-Gulf War military service were risk factors for psychiatric hospitalization, while two clinics ("clinic C" and "clinic G" in Fig 2) were protective factors. Our HTE analysis found the strongest evidence of HTE for the control cohort among those with a prior psychiatric hospitalization in the previous year, preferentially reducing psychiatric hospitalization by a factor of nearly five. We observed evidence for CCM preferentially reducing psychiatric hospitalization rates in patients with depression and those with a single medical diagnosis (but not more than one).

## Interpretation

**Trajectory of CCM effect.** We found that the change in magnitude of CCM effect on psychiatric hospitalization began immediately after CCM implementation support. Despite the baseline trend of higher psychiatric hospitalization rates in the CCM cohort, by the second month the entire range of our estimate no longer included the null. The increasing CCM effect magnitude gradually slowed but did not reach steady state by the end of the first year. The time course of acute care utilization effect is consistent with the broad parameters of effects seen in CCM controlled trials [4,5], and is a gradual process of learning and system change [19,20]. Additionally, a recent study suggests that time course of CCM fidelity may play a role in predicting clinical outcomes. Specifically, an effectiveness trial analysis of CCM in 666 U.S. military members examined the relationship between CCM fidelity and symptom reduction for depression and PTSD [21]. While they did not use survival analysis, they approximated fidelity and symptom trajectories with four assessments over one year. They found that trajectories which included high CCM fidelity later in the course, especially when sustained, was most associated with reduction of depression and PTSD symptoms. Again, while our study did not specifically track CCM fidelity, the trajectory of CCM's effect on psychiatric hospitalization followed a time course that is congruent with an effect that maximizes later and is sustained throughout. This suggests that while it might be reasonable to assess impact on psychiatric hospitalization when the effect size begins to stabilize, around six months, confirming an association between CCM fidelity may be better assessed later in the course of implementation, closer to one year. Further implementation trials may also wish to extend beyond one year to examine whether the CCM effect on acute psychiatric hospitalization continues to grow.

**Differential effect of prior psychiatric hospitalization.** Having a history of psychiatric hospitalization in the prior year favored usual care over CCM (RHR 4.92; CI 3.15–7.7). If this new finding is replicated it could conflict with the interpretation that CCM might lower acuity for more complicated patients [18]. However, it is also worth considering contextual factors related to psychiatric hospitalization. Among patients with more recurrent acute presentations, acute psychiatric hospitalization may not always reflect a 'poor' patient outcome. Indeed, there is current discussion about what constitutes appropriate psychiatric admissions [22], similar to the discussion of ambulatory care-sensitive admissions for chronic medical conditions [23]. Future analyses may consider other outcomes for which psychiatric admission is indicated to prevent, (e.g., loss of job, relationships, or life).

**Differential effect on patients with depression.** Systematic reviews and meta-analyses have previously found clinical and economic benefits of CCM across a broad range of individual mental health conditions [4,5]. To the best of our knowledge, past studies have not assessed an effect of CCM specifically on rates of psychiatric hospitalization. However, to the extent that reduced psychiatric hospitalizations represent clinical benefit, our results suggest preferential benefit of CCM among patients with depression (RHR 0.61 CI 0.41–0.91). This finding is consistent with a 2016 registry-based retrospective survival analysis of CCM for depression in a primary care setting [24] that found that time until remission was nearly two and half

times shorter than patients receiving usual care. We are aware of only one set of meta-analyses that examined effects of the CCM across different mental health diagnostic groups [4,5]. These analyses did not find a preferential response in studies of depression, although that diagnosis was represented by the most trials. However, these clinical trials, and a more recent study of individuals with bipolar disorder [25] utilized consented samples rather than the whole-population approach taken in our analyses. Our trial thus included patients with greater heterogeneity, including both less and more severely symptomatic individuals than are typically represented in clinical trials. Additional population-based studies are needed to confirm this finding.

**Inverted U-function medical morbidity preferential effect.**   We report the unexpected finding that, compared to those without any medical morbidity, only those with a single morbidity, but *not* those with more than one medical morbidity, showed a preferential response for CCM reducing psychiatric hospitalization (RHR 0.52; 95% CI 0.31–0.86). Our unique approach to using acute psychiatric hospitalization data makes direct comparison to past literature complicated. Perhaps having a single medical morbidity might mean a patient is better connected to primary care and may have more relevant clinical points of contact; there may also be a ceiling effect in which further medical complexity disrupts routine psychiatric care. Future research may wish to model psychiatric hospitalization across a greater range of comorbidity counts (or types) to better establish this relationship.

## Limitations

**Unmeasured confounding.**   The quasi-experimental study design limited the randomization of trial arm status to the level of cluster. Thus, residual unmeasured confounding from latent variable(s) could exist, potentially altering our results if the magnitude of unmeasured confounding were large enough. However, our sensitivity analysis identified the parameter values that would be required to explain away CCM's effect on psychiatric hospitalization due to unmeasured confounding. Specifically, we found that: (i) The magnitude of all unmeasured confounding would have to be equivalent to a HR *greater than* 2.4. (ii) As a corollary, any unmeasured confounding less than a HR of 2.4 could not invalidate our findings. (iii) Additionally, these associations would have to be exerted *independently of* all measured covariates specified in the extended model. Using a stricter threshold, for a confounded latent variable to move the upper bound of our estimate to include the null, the above criteria would require a magnitude of association equivalent to a HR greater than 1.7. Finally, sensitivity analysis using E-values are only interpretable in the context of all efforts to account for confounding, especially among the measured covariates [17].

Considering that we addressed confounding bias at both the level of design (e.g., site balancing) and level of analysis (e.g., multilevel modeling) the calculated E-values support the robustness of our findings.

**External validity.**   Our study was limited to United States veterans. Compared to the civilian population, veterans have higher rates of mental health conditions, particularly PTSD, as well as higher rates of general medical chronic disease [26]. Additionally, we could not provide estimate of risk contribution or subgroup HTE analysis for female sex. However, we were able to adjust for specific psychiatric diagnoses, as well as for medical comorbidities. And we were able to adjust other covariates for sex in our analyses.

**Limitations of subgroup analyses.**   Our approach to analyzing HTE, essentially a second differences approach generalized for probability ratios, has several limitations [16,27]: (i) Efficiency; statistical power is sacrificed when the stratified models were subset by trial arm, compounding the fact the larger trial was not powered for these subgroups. The absence of

evidence for HTE for any subgroup should thus not be interpreted as evidence in support of uniformity of treatment response. (ii) Linearity; assessing HTE individually for each subgroup ignores heterogeneity *within patients*. Non-linear interactions between variables, such as the interactions between different psychiatric comorbidities or social demographics or more complex systems type interactions will not be captured. (iii) Risk magnification; because the baseline absolute risk of the CCM group was higher than the control group, it gives an 'advantage' to CCM for reducing psychiatric hospitalizations when the main measure of association is relative, such as a hazard ratio. However, the baseline risk for CCM included a range that also included the null and was not many times greater than controls. (iv) Multiplicity; multiple exploratory comparisons increase chances of false positive findings. As a result, we have chosen to not highlight HTE trends. Our HTE findings' intended use is strictly limited to future prospective confirmatory studies that can improve upon efficiency and capture potentially complex interactions within individuals. This might be best accomplished with newer methods such as Bayesian, nonparametric, and predictive risk models [27,28]. Advances in making ensemble methods available to researchers have also been developed for HTE using time-to-event data [29].

**Secular trends and intervention-time effects.** Staggered start times are inherent to trials with stepped wedge designs, and may threaten validity if not adjusted for in survival analysis [30]. However, unlike most stepped wedge designs, our study featured a concurrent control group that never received the intervention, mitigating these concerns. For example, *within waves*, both CCM and controls were observed contemporaneously at the same medical centers. Any secular trends would have influenced both cohorts at the same time. Additionally, *between waves*, time-to-event data was set according to the study protocol time, such that follow-up time was always relative to the number of days since start of CCM implementation support. As a result, any variation in intervention effect over time (e.g., a lag time) would be compared equally in the analysis regardless of calendar start time. Lastly, clinic site was specified in all models to assess for potential heterogeneity. However, patients from a given clinic also share the same start time, which conditions the models on time (yet limiting its ability to purely capture clinic site heterogeneity).

**Mapping CCM fidelity to treatment effect.** These analyses investigated care utilization (hospitalization rates) independent of any process measures of implementation. While we cannot determine how inpatient utilization might map onto CCM fidelity, meta-analysis had previously found that lower CCM fidelity was associated with smaller and more variable pooled effect sizes [31]. As periods leading up to psychiatric hospitalization generally correspond with clinical decompensation, to the extent that higher CCM fidelity might predict lower clinical acuity, we might reasonably anticipate that higher CCM fidelity would also correspond with reductions in psychiatric hospitalization.

## Significance for health care systems, quality improvement, and related policies

The development of *learning health care systems* is a consensus goal of the National Academy of Medicine (NAM) (previously the Institute of Medicine) intended to "ensure innovation, quality, safety, and value in health care" [32]. Two noncomprehensive features of a learning health care system are (i) the capacity to synthesize the best evidence available for policies that maximally improve the quality of care, especially those that improve collaborative care practices and (ii) application of embedded processes to facilitate the collection and use of health informatics (among other factors) in real-time for continuous assessment and improvement of health care quality. The primary implementation trial this study was based on exemplified the

first feature, resulting in VA wide policies based on evidence synthesis for CCM in a general mental health patient population; additionally, the second feature was demonstrated in follow-up work with facilitators in VA's Transformational Coach program (known as 'T-Coaches') intended to expand capacity for facilitated implementation support in service of scaling CCM to VA sites outside the implementation trial [33].

In the years following their seminal report, NAM published ten recommendations towards achieving a learning health care system [20]. The first of these calls for a digital infrastructure allowing for a continuous collection of health informatics (including outcomes of care). The VA's CDW already has the capacity to capture health outcomes, such as psychiatric hospitalization. And this information could support the early identification of problems during further scaling of CCM. Importantly, early warnings for sup-performing CCM sites must be timely. All QI activities promote rapid cycle testing, which is a reason why the capacity for near real-time performance metrics are a core feature of learning health care systems. As such, the utility of CDW data for this purpose is limited without a near continuous reference for the expected real-world performance. Our time course simulation of CCM's effect on psychiatric hospitalization rates bridges this gap by providing a performance reference for each day following implementation start. This contribution is aligned with the ninth NAM recommendation, intended to support the environment for policy change in a learning health care system. Specifically, the ninth recommendation calls for health care delivery organizations to increase transparency of system performance, including care outcomes, in part to guide improvement efforts. To be clear, greater transparency surrounding expected performance of CCM primarily improves screening. Sites found to miss expected performance benchmarks will require additional assessment to identify the underlying causes and the specific implementation level interventions needed.

## Conclusion

To our knowledge, this is the first survival analysis of CCM effects on psychiatric hospitalization rates, providing critical information on both the time-course of CCM effectiveness and HTE. Overall, we found large variation across both time and subgroups, but on average CCM reduced rates of psychiatric hospitalization by 17% to 49%; this average effect began immediately on commencement of CCM implementation support and grew in effect size through the final study day, one year later. The modeled trajectory suggests that assessments of CCM effect on psychiatric hospitalization would be well-timed after the change in effect magnitude begins to flatten out, as early as six months post-implementation support start. Our HTE findings should be cautiously interpreted but will be of use to future researchers interested in the utility of the CCM in various subpopulations.

## Acknowledgments

The authors wish to acknowledge contributions from members of our CCM research team, and to Garrett Fitzmaurice Sc.D. and Robert Lew Ph.D. for consulting on aspects of our statistical plan.

## Author Contributions

**Conceptualization:** Michael A. Ruderman, Mark S. Bauer.

**Data curation:** Michael A. Ruderman, Kelly Stolzmann, Mark S. Bauer.

**Formal analysis:** Michael A. Ruderman.

**Funding acquisition:** Bo Kim, Mark S. Bauer.

**Investigation:** Michael A. Ruderman, Bo Kim, Mark S. Bauer.

**Methodology:** Michael A. Ruderman.

**Project administration:** Michael A. Ruderman, Bo Kim, Mark S. Bauer.

**Resources:** Bo Kim, Kelly Stolzmann, Mark S. Bauer.

**Software:** Michael A. Ruderman.

**Supervision:** Bo Kim, Christopher J. Miller, Mark S. Bauer.

**Validation:** Michael A. Ruderman.

**Visualization:** Michael A. Ruderman.

**Writing – original draft:** Michael A. Ruderman, Bo Kim, Mark S. Bauer.

**Writing – review & editing:** Michael A. Ruderman, Bo Kim, Kelly Stolzmann, Samantha Connolly, Christopher J. Miller, Mark S. Bauer.

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
