## [Decision Letter · Decision Letter 0]

19 Oct 2020

PONE-D-20-19306

Time Course and Heterogeneity of Treatment Effect of the Collaborative Chronic Care Model on Psychiatric Hospitalization Rates: A Survival Analysis Using Routinely Collected Electronic Medical Records

PLOS ONE

Dear Dr. Ruderman,

Thank you for submitting your manuscript to PLOS ONE. After careful consideration, we feel that it has merit but does not fully meet PLOS ONE’s publication criteria as it currently stands. Therefore, we invite you to submit a revised version of the manuscript that addresses the points raised during the review process.

We look forward to receiving your revised manuscript.

Kind regards,

Kyoung-Sae Na, M.D.

Academic Editor

PLOS ONE

Journal Requirements:

2. In the ethics statement in the manuscript and in the online submission form, please provide additional information about the patient records used in your study.

Specifically, please ensure that you have discussed whether all data were fully anonymized before you accessed them and/or whether the IRB or ethics committee waived the requirement for informed consent.

If patients provided informed written consent to have data from their medical records used in research, please include this information.

Reviewers' comments:

Reviewer's Responses to Questions

**Comments to the Author**

1. Is the manuscript technically sound, and do the data support the conclusions?

Reviewer #1: Yes

Reviewer #2: Yes

2. Has the statistical analysis been performed appropriately and rigorously? 

Reviewer #1: Yes

Reviewer #2: Yes

3. Have the authors made all data underlying the findings in their manuscript fully available?

Reviewer #1: Yes

Reviewer #2: Yes

4. Is the manuscript presented in an intelligible fashion and written in standard English?

Reviewer #1: Yes

Reviewer #2: Yes

5. Review Comments to the Author

Reviewer #1: In this article, the authors used survival analysis to explore time course and heterogeneity of CCM’s effect on psychiatric hospitalization rates across demographic, clinical, and site level characteristics, as well as the overall variation of treatment effect over time. The paper is interesting. However, I have some question that should be addressed to improve it.

Commends:

1.The authors should state clearly that why use two Cox models subset by trial arm to estimate covariate risk in lines 248 to 254. The rational can be from Fig. 1, since the CCM effect on psychiatric hospitalization is not constant over time, a single Cox regression model is not appropriate for the trial arms.

2.The authors fit two Cox model by trial arm to evaluate the covariate risk and further estimate the RHR and the corresponding 95% CI using simulation method.

The author may also consider a stratified Cox model to estimate the RHR.

3.From the results in Fig. 3, we observe evidence for CCM preferentially reducing psychiatric hospitalization rates in patients with depression and those with a single medical diagnosis. Are the results similar when removing the non-significant covariates and fit a smaller model and also consider any interaction effect between the covariates?

Reviewer #2: This analysis has been carried out on a large number of patients and provides interesting information on the the course and heterogeneity of the CCM effect on reducing psychiatric hospitalisations. To my knowledge, statistical methods used appear sound. The authors might want to expand on the clinical and policy-level significance of the findings.

6. PLOS authors have the option to publish the peer review history of their article (what does this mean?). If published, this will include your full peer review and any attached files.

Reviewer #1: No

Reviewer #2: No

---

## [Author Response · Author response to Decision Letter 0]

16 Feb 2021

Response to Editor’s and Reviewers’ Comments

Journal Requirements:

Editor Comment 1: Please ensure that your manuscript meets PLOS ONE's style requirements, including those for file naming.

Response:

-Title has been changed from title case to sentence case (page 1).

- Affiliations of the corresponding author updated to reflect change in institutions (page 1).

-Level headings changed from capitalized case to sentence case, font sizes changed to reflect heading levels (page 1).

-References have been adjusted (pages 32-35).

-Rechecked that file naming conventions were consistent with PLOS guidelines.

-Rechecked that figure images conform to journal specifications using PACE.

Editor Comment 2: In the ethics statement in the manuscript and in the online submission form, please provide additional information about the patient records used in your study. Specifically, please ensure that you have discussed whether all data were fully anonymized before you accessed them and/or whether the IRB or ethics committee waived the requirement for informed consent. If patients provided informed written consent to have data from their medical records used in research, please include this information.

Response: An ethics statement has been added (lines 132-141). The statement includes information about whether data were anonymized and information about the IRB exemption.

Editor Comment 3: We note that you have stated that you will provide repository information for your data at acceptance. Should your manuscript be accepted for publication, we will hold it until you provide the relevant accession numbers or DOIs necessary to access your data. If you wish to make changes to your Data Availability statement, please describe these changes in your cover letter, and we will update your Data Availability statement to reflect the information you provide.

Response: We would like to change our data availability statement. We have included a revised statement in our cover letter. Specifically, we have provided a new contact to establish a data use agreement, CHOIR@va.gov. This email account is not linked to any individual person or author. This account will be monitored for requests made by anyone wishing to access our data, even after the current authors’ service ends. There are protocols in place such that persons monitoring the account will be able to initiate data use agreements and locate the data to send to requestors, even if they are without prior knowledge of the study.

Response to Reviewer #1 comments:

Author note: Below we reference line numbers to help direct reviewers and editors to specific revisions in the manuscript. Please be aware these line numbers all refer to the track changes version of the manuscript.

R1 Comment 1. The authors should state clearly that why use two Cox models subset by the trial arm to estimate covariate risk in lines 248 to 254 [author note: this corresponds to lines 291-299 of the revised manuscript with track changes]. The rational can be from Fig. 1, since the CCM effect on psychiatric hospitalization is not constant over time, a single Cox regression model is not appropriate for the trial arms.

Response: In response to this comment, we have clarified in lines 264-271, lines 292-305, and lines 407-409 of the track changes revised version of the manuscript. These revisions describe the differences between modeling time-varying effects versus modeling covariate effects and related heterogeneity of treatment effects.

Specifically, this manuscript presents the findings from three models, an extended cox model and two stratified cox models (each subset by trial status). The extended model is included to simulate time-varying treatment effects, and the two stratified cox models are used to generate ratios of hazard ratios.

Fig. 1 presents the time-varying treatment effects simulated from the extended model only. There are no time varying effects modeled in either of the other two models (the two stratified models that are subset by trial status). We agree that a hypothetical single cox model (including subjects from both trial status arms) would be inappropriate, due to variation in effect over time.

Importantly, trial status itself is one of the variables that violated the proportional hazards assumption. Using two stratified models, subset by trial status, the variable is effectively removed from each model (more specifically, it is not specified in the model). We conducted the diagnostics discussed in the manuscript to both subset cox models. Based on these diagnostics, even after being subset, two other variables besides trial status remained in violation of the proportional hazards assumption. To adjust for the remaining violators, we applied a stratified procedure to the subset cox models, yielding the two stratified models (also described in the original manuscript). 

After this procedure, further diagnostics revealed there were no remaining variables in violation of the proportional hazards assumption. We also described the trade-off of using a stratified procedure (specifically, that while the violators can be adjusted for in the effect estimates of non-violators, explicit estimates of the effects for the two remaining violators is not possible). Consequently, no effect measures for the two violators are included with the forest plot of covariate effect estimates in fig 2 (nor corresponding ratios of hazard ratios in fig 3). Throughout the revised manuscript (264-271, lines 292-305, and lines 407-409) we now provide improved descriptions of this rationale.

R1 Comment 2. The authors fit two Cox model by trial arm to evaluate the covariate risk and further estimate the RHR and the corresponding 95% CI using simulation method. The author may also consider a stratified Cox model to estimate the RHR. 

Response: Thank you for this suggestion! We realize now that our wording in the original submission was unclear on some details related to our Cox models. We wish to clarify a few points:

First, to address ambiguity about whether a standard Cox model was used, we have added additional language in two sections (lines 49-50, 292-295) clarifying that no standard Cox proportional-hazards models were used to estimate either the time-course or heterogeneity of treatment effect. The two stratified Cox models (presented in Fig. 2) were used to calculate the RHRs (as presented in Fig. 3). There are no standard Cox models used in any of our analyses (for time trajectory or heterogeneity of treatment effect). We believe our revisions on lines 49-50 and 292-295 remove ambiguity concerning use of ‘extended’ and ‘stratified’ variants of the Cox models.

Second, to address ambiguity about which models were simulated and which weren’t, we have added explicit statements (lines 47-49) that we do not employ simulation to estimate either the point estimates or confidence intervals for the covariate risk contributions or their subsequent RHRs. We use the hazard ratio point estimates from the stratified Cox model (the CCM subset) and divide it by the hazard ratio point estimate for the second stratified Cox model (the control subset). The confidence intervals for the RHRs are calculated from the standard errors associated with the respective hazard ratios’ point estimates. These point estimates and standard errors are calculated in the usual way (for 95% confidence intervals). Simulation is only used to estimate (and visualize) the instantaneous hazard ratio estimates and the instantaneous confidence intervals estimates for the time varying effects in the extended models only. We believe the addition of explicit statements that the stratified models and RHR were not simulated will eliminate ambiguity.

The rationale for not using a single stratified model is as follows. By definition, the ratio of hazard ratios is a second differences approach to heterogeneity of treatment effect. You can think of the hazard ratios from each subset model as being the ‘first difference.’ The hazard ratio for one subset model becomes the numerator and the other hazard ratio for the second subset model becomes the denominator, the ratio of which describes the ‘second difference’ (or the ratio of the two hazard ratios). Said another way, the relative effect from a model in one sample is compared, relatively, to the relative effect of the same model in a different sample. For this study, that means the RHR is defined as the hazard ratio of one stratified model (subset to the CCM group only), divided by the hazard ratio of the same stratified model (but subset to the control group only). In this way, the relative effect of the same model in two different samples can be compared (by second differences).

Based on statistical guidance from both our research colleagues and the Harvard Clinical and Translational Science Center, we chose against modeling second differences with varying interaction terms in a single model because: (i) using a single stratified model on one sample (combining CCM and control groups) would re-introduce the trial status variable into the models, which violated the proportional hazards assumption, (ii) we find the ratio of hazard ratios is more readily interpretable than the coefficients of interaction terms, and (iii) it does not offer less biased modeling of heterogeneity of treatment effects.

R1 Comment 3. From the results in Fig. 3, we observe evidence for CCM preferentially reducing psychiatric hospitalization rates in patients with depression and those with a single medical diagnosis. Are the results similar when removing the non-significant covariates and fit a smaller model and also consider any interaction effect between the covariates?

Response: We thank reviewer 1 for this comment. We have addressed this specifically by adding language on lines 404-409 describing that effect estimates for covariate risk contributions and RHRs were overall robust when we excluded non-significant variables or included selected covariate interactions. 

In addition, we realized we did not originally provide a description of our modeling process. To address this, we have added a paragraph (lines 219-225) in the now re-titled section, “Modeling and analytic overview.” This addition includes additional information about the process we used to specify of model candidates and how we selected among them.

Response to Reviewer #2 comments: 

R2 Comment 1: The authors might want to expand on the clinical and policy-level significance of the findings.

Response: Thank you for this helpful suggestion to strengthen the manuscript by expanding on our findings’ significance. To address this comment, we have included an additional discussion section (lines 606-640) called, “Significance for health care systems, quality improvement, and related policies.” We include discussion about VA nationally adopting the CCM model, including scaling to additional sites beyond those included in the implementation trial. In this section we integrate our findings into the broader goals of the National Academy of Medicine to develop ‘learning health care systems.’

Additional revisions not prompted by reviewer comments: As we made revisions, we made additional changes that serve to simplify language for clarity and understanding. Specifically, we improved the readability of the section on unmeasured confounding (lines 532-545). We also mad minor corrections throughout the revised manuscript, primarily of a typographical nature.

---

## [Decision Letter · Decision Letter 1]

10 Mar 2021

Time course and heterogeneity of treatment effect of the collaborative chronic care model on psychiatric hospitalization rates: A survival analysis using routinely collected electronic medical records

PONE-D-20-19306R1

Dear Dr. Ruderman,

We’re pleased to inform you that your manuscript has been judged scientifically suitable for publication and will be formally accepted for publication once it meets all outstanding technical requirements.

Kind regards,

Kyoung-Sae Na, M.D.

Academic Editor

PLOS ONE

Additional Editor Comments (optional):

Reviewers' comments:

Reviewer's Responses to Questions

**Comments to the Author**

1. If the authors have adequately addressed your comments raised in a previous round of review and you feel that this manuscript is now acceptable for publication, you may indicate that here to bypass the “Comments to the Author” section, enter your conflict of interest statement in the “Confidential to Editor” section, and submit your "Accept" recommendation.

Reviewer #1: All comments have been addressed

2. Is the manuscript technically sound, and do the data support the conclusions?

Reviewer #1: Yes

3. Has the statistical analysis been performed appropriately and rigorously? 

Reviewer #1: Yes

4. Have the authors made all data underlying the findings in their manuscript fully available?

Reviewer #1: Yes

5. Is the manuscript presented in an intelligible fashion and written in standard English?

Reviewer #1: Yes

6. Review Comments to the Author

Reviewer #1: (No Response)

7. PLOS authors have the option to publish the peer review history of their article (what does this mean?). If published, this will include your full peer review and any attached files.

Reviewer #1: No

---

## [Editor Report · Acceptance letter]

17 Mar 2021

PONE-D-20-19306R1 

Time course and heterogeneity of treatment effect of the collaborative chronic care model on psychiatric hospitalization rates: A survival analysis using routinely collected electronic medical records 

Dear Dr. Ruderman:

I'm pleased to inform you that your manuscript has been deemed suitable for publication in PLOS ONE. Congratulations! Your manuscript is now with our production department. 

Kind regards, 

on behalf of

Dr. Kyoung-Sae Na 

Academic Editor

PLOS ONE